# Targeting Melanin in Melanoma with Radionuclide Therapy

**DOI:** 10.3390/ijms23179520

**Published:** 2022-08-23

**Authors:** Kevin J. H. Allen, Mackenzie E. Malo, Rubin Jiao, Ekaterina Dadachova

**Affiliations:** College of Pharmacy and Nutrition, University of Saskatchewan, Saskatoon, SK S7N 5E5, Canada

**Keywords:** metastatic melanoma, radioimmunotherapy, melanin, targeted radionuclide therapy, benzamides, nicotinamides, picolinamides, clinical trials, B16F10 melanoma, dosimetry

## Abstract

Nearly 100,000 individuals are expected to be diagnosed with melanoma in the United States in 2022. Treatment options for late-stage metastatic disease up until the 2010s were few and offered only slight improvement to the overall survival. The introduction of B-RAF inhibitors and anti-CTLA4 and anti-PD-1/PD-L1 immunotherapies into standard of care brought measurable increases in the overall survival across all stages of melanoma. Despite the improvement in the survival statistics, patients treated with targeted therapies and immunotherapies are subject to very serious side effects, the development of drug resistance, and the high costs of treatment. This leaves room for the development of novel approaches as well as for the exploration of novel combination therapies for the treatment of metastatic melanoma. One such approach is targeting melanin pigment with radionuclide therapy. Advances in melanin-targeting radionuclide therapy of melanoma can be viewed from two spheres: (1) radioimmunotherapy (RIT) and (2) radiolabeled small molecules. The investigation of mechanisms of the action and efficacy of targeting melanin in melanoma treatment by RIT points to the involvement of the immune system such as complement dependent cytotoxicity. The combination of RIT with immunotherapy presents synergistic killing in mouse melanoma models. The field of radiolabeled small molecules is focused on radioiodinated compounds that have the ability to cross the cellular membranes to access intracellular melanin and can be applied in both therapy and imaging as theranostics. Clinical applications of targeting melanin with radionuclide therapies have produced encouraging results and clinical work is on-going. Continued work on targeting melanin with radionuclide therapy as a monotherapy, or possibly in combination with standard of care agents, has the potential to strengthen the current treatment options for melanoma patients.

## 1. Introduction

Melanoma is a cancer that originates in the melanocytes, the melanin pigment-producing cells. Melanoma constitutes the fifth most common form of cancer with an estimated 99,780 people in the United States expected to be diagnosed in 2022 [1,2]. The early detection of melanoma is associated with remarkable prognosis, but the diagnosis of late-stage disease is still associated with poor survival. The approval of BRAF-targeted therapies and immunotherapy and their subsequent rapid incorporation into standard of care resulted in a decline of nearly 8% in mortality rates across all stages of melanoma in the U.S. between 2013 and 2019 [3].

Immunotherapy treatments have focused on two main targets: cytotoxic T-lymphocyte associated protein 4 (CTLA-4), and the receptor–ligand pair, programmed cell death-1 (PD-1) and programmed death-ligand 1 (PD-L1) [4]. These treatments alter T-cell regulation, although their spatial and temporal effector function indicates that they do so through different pathways. CTLA-4 is targeted using the antibody ipilimumab and can result in an extended survival [5]. PD-1/PD-L1 are targeted with pembrolizumab and nivolumab, and both improve the overall survival equally [6,7]. Vemurafenib, the first targeted BRAF inhibitor, was initially approved by the FDA in 2011, followed by the approval of darafenib and encorafenib, two additional inhibitors. These therapies targeted mutated BRAF, present in 40–60% of patients with melanoma, and improved the overall survival to 50% at 17 months, however, these results are typically followed by relapse [8,9,10]. In 2018, the FDA approved the use of combination therapy of the BRAF inhibitor encorafenib with the targeted MEK inhibitor, binimetinib, after the COLUMBUS Trail showed a doubling in progression-free survival in the combination therapy relative to monotherapy [11].

Despite new revolutionary treatments and their significant impact on improving 5-year survival rates for metastatic melanoma, there are concerning side effects associated with immunomodulatory therapy, and BRAF-targeted therapy is associated with eventual drug-resistance. There is a significant need for alternative approaches to treat metastatic melanoma that can be used as monotherapies or in combination with standard of care therapies, and which can evade the eventual development of resistance.

The development of therapeutics in cancer research are often focused on exploiting features of the malignancy that differentiate it from healthy tissue. In the case of melanoma, the pigment melanin has historically been utilized due to its presence in >92% of melanomas [12,13]. The cutaneous melanin pigment is known to perform many functions with the protection against the hazardous effects of solar radiation being one of the most important for humans [14]. As our skin is exposed to various levels of UV radiation, melanogenic activity is regulated accordingly via unique molecular sensors and signal transducers. The multiplicity of melanin’s functions makes it necessary for melanogenesis to be a highly structurally controlled process on a cellular level, which includes the ability to utilize multiple melanin precursors [14]. Early studies recognized that melanin precursor analogues such as ^10^B-p-boronophenylalanine (BPA) could be targeted to rapidly growing melanomas with high melanin production, allowing for the delivery of a dose of mixed linear energy transfer (LET) radiation from boron neutron capture therapy (BNCT) [14,15,16,17,18]. Using melanin as a target is ideal due to its ubiquitous nature in melanoma, but considerations must also be made for the impact that the pigment has on melanocyte behavior, the development of malignancy, and response to therapy [19,20,21,22,23,24].

There are two possible ways to target melanin in melanoma—one would be to use radiolabeled small molecules that are able to cross the cellular membrane and bind to melanin in the melanosomes. The potential downside of this approach is that such compounds might be also crossing the membranes of the healthy melanized cells such as in the retina of an eye as well as in macrophages, which have phagocyted melanin. The second approach is to use non-internalizing antibodies or peptides to melanin. These radiolabeled antibodies would target the melanin released from rapidly dividing melanoma cells while viable tumor cells nearby would be killed by radiation emitted by the radiolabeled molecules (so called “cross-fire effect”). Targeting extracellular melanin would also spare healthy melanized tissues due to the non-internalizing nature of the antibodies. As the monomer units of human eumelanin (5,6-dihydroxyindole and 5,6-dihydroxyindole-2-carboxylic acid) and of pheomelanin (benzothiazine) have many structural similarities—small molecules and antibodies alike should be able to recognize and bind both types of melanin in the tumors.

It should be noted that melanin is known for its radiation protective properties [25,26,27,28,29,30,31], but disfunction in the regulation of melanogenesis contributes to melanoma initiation and development [19]. Increased melanin production in the context of melanoma is associated with altered cellular metabolism [20] and an overall reduction in survival times [24]. Furthermore, elevated melanogenesis alters the effectiveness of external beam radiation therapy (EBRT) in melanoma patients, and melanoma cell culture studies suggest that the inhibition of melanogenesis could improve radiosensitivity [23,32]. Understanding melanin’s role in melanoma induction and progression helps in the development and optimization of new and current therapeutics. This scope of this review was to highlight the most promising developments in the last 10 years in the radionuclide treatment of melanoma targeting melanin.

## 2. Methodology

The PubMed and Web of Science databases were searched, with the years of search being 2011–2022. The search terms were melanoma, targeted radionuclide therapy, melanin, radioimmunotherapy and clinical trials, and they were used in various combinations. The exclusion term was α-melanocyte-stimulating hormone (α-MSH). The references older than 2011 were included only if required to provide the background information on melanin or the targeting molecules.

## 3. Targeting Melanin Antigen in Melanoma with Radiolabeled Antibodies

Historically radioimmunotherapy (RIT) targets cell surface antigens due to their accessibility, which renders intracellular targets out of reach for RIT. Despite this being the case for healthy growing cells, the high mitotic activity of cancer cells and significant necrotic death could result in the release of intracellular melanin into the extracellular space. Melanin became our antigen of interest in melanoma due to its substantial presence in the tumors and its stability upon release into the extracellular space, thus ensuring the integrity of the target. By targeting melanin released from rapidly dividing melanoma tumors with radiolabeled antibodies for RIT, we aimed to deliver cytotoxic radiation to these cells and disrupt tumor growth [33]. This method deviates from traditional cell killing via the direct bombardment of cells with radiation by exploiting a “cross-fire” effect to adjacent tumor cells. The benefit of targeting extracellular melanin is that healthy cells expressing melanin cannot be targeted, minimizing the off-target effects, resulting in an excellent safety profile. The efficacy of EBRT in the treatment of melanoma is closely tied to the amount of melanin present [23], likely due to the ability of melanin to scavenge free radicals [34,35,36] generated by the therapy. Since RIT is delivered directly to the tumor site—it is able to deposit high LET radiation to the tumor cells, employing both the direct and indirect cytocidal effects of radiation [37] and hopefully minimizing the impact of melanin’s scavenging capacity to neutralize reactive free radicals. In order to explore this hypothesis, we utilized an IgM antibody, 6D2, which was isolated from mice immunized with a fungal eumelanin [38], and performed an extensive pre-clinical development and evaluation before embarking on a clinical trial [39]. In addition, computer-simulated tumor dosimetry demonstrated that ^188^Re-6D2 could deliver tumoricidal doses to tumors within the wide range of melanin concentrations (up to 100 less melanin than in the primary tumors) [40].

A clinical, imaging-focused study was conducted using rhenium-188 (^188^Re) labeled to 6D2 in individuals with stage IV metastatic melanoma who had failed the standard of care therapy [39]. Whole-body SPECT/CT via planar scintigraphy was performed and a time dependent distribution was determined. The analysis showed a significant uptake of ^188^Re-6D2 in most metastatic lesions, with no indications of dose limiting toxicities [41]. The promising results from the first investigatory study allowed for expansion into Phase 1a and Phase 1b clinical trials (NCT00399113 and NCT00734188) [42]. The goals of the Phase 1 clinical trials were to establish the pharmacokinetics of ^188^Re-6D2 and to evaluate toxicity with dose escalation. Patient follow-ups were conducted 2- and 6-weeks post-treatment, and patients in the phase 1b trial returned every 8 weeks until the progression of melanoma was observed. Reassuringly, analysis of the whole-body planar scintigraphy showed no uptake in the healthy melanized tissues such as the retina and skin, while showing that ^188^Re-6D2 was localizing in the melanoma tumors and metastases. Post study analysis showed that the treatment resulted in a 4.5 month extension of life (13 month median survival, up from 8.5 month historical median survival of the same stage melanoma patients treated with the standard of care therapies at Hadassah Medical Center where the trial was conducted). Furthermore, no hematological side effects were observed with only mild toxicities. Due to the limitations of the ^188^W/^188^Re generator at the time of the study, it was not possible to determine the maximum tolerated dose. This work supported the use of melanin as a specific and effective target for RIT and demonstrated the utility of RIT in extending survival in metastatic melanoma.

To further understand the mechanism of action of the ^188^Re-6D2 RIT, we investigated the role of antibody-dependent cell-mediated cytotoxicity (ADCC) and compliment-dependent cytotoxicity (CDC) in fighting melanoma using the A2058 mouse melanoma model [43]. During the labeling process, the molar amount of the radioisotope can be 100-fold less than that of the antibody present, resulting in a significant portion of an injected dose of the RIT containing a non-radioactive antibody. To investigate whether this “cold” antibody was involved in ADCC or CDC, we injected an unlabeled 6D2 antibody into mice bearing A2058 melanoma tumors. No evidence of ADCC was observed, which was expected, as this is not a typical feature of IgM antibodies, which are, however, known to initiate CDC [44]. The assay used to measure CDC showed a significant 6D2-dependent CDC in comparison with a no mAb control, indicating a complement-dependent component to the tumoricidal activity of ^188^Re-6D2.

Drug resistance hinders the efficacy of current first line therapy for the treatment of metastatic melanoma, which leads to poor prognoses for patients. Both the approved BRAF inhibitors and the immunomodulating antibodies are prone to such resistance with melanoma stem cells (MSC) being implicated in the development of such resistance [45]. We investigated whether ^188^Re-6D2 was prone to a similar resistance paradigm using the A2058 murine melanoma model [46]. Tumors were probed for the presence of MSC markers (chemoresistance mediator ABCB5 and H3K4 demethylase JARID1B), and no significant difference was observed between the RIT treatment groups relative to the control groups, indicating that ^188^Re-6D2 administration is not likely subject to the MSC-dependent drug resistance that plagues traditional therapy.

Due to the technical challenges of working with an IgM antibody, all further work in targeting melanin for the treatment of melanoma with RIT by our group was focused on the development and evaluation of a novel murine anti-melanin IgG antibody—8C3 [47]. IgG antibodies can be easily humanized, allowing for the development of Good Manufacturing Practice (GMP) grade antibodies that could conceivably be utilized in clinical trials. Comparative structural analysis showed significant differences between the 6D2 IgM and 8C3 IgG antibodies, indicating that the antibodies likely interact with different melanin epitopes [47]. Despite this difference, the in vivo evaluation of the radiolabeled 8C3 in the murine B16-F10 melanoma model indicated effective tumor targeting, with no interaction with melanin in the healthy cells. Moreover, ^213^Bi- and ^188^Re-8C3 RIT demonstrated improved efficacy in reducing the metastatic melanoma lesions in the lungs in comparison to that of 6D2 RIT, and the 8C3 IgG allowed for efficient radiolabeling, requiring minimal post-labeling purification, unlike the 6D2 IgM [47]. In transitioning to a humanized anti-melanin IgG, an evaluation of the 8C3-derived chimeric antibody pharmacokinetics was performed [48], followed by the evaluation of a fully humanized 8C3 (h8C3) anti-melanin IgG [49].

A comparative study of ^213^Bi- versus ^177^Lu-h8C3 in the B16-F10 murine model presented promising results for the fully humanized anti-melanin IgG mAb [49]. Biodistribution, microSPECT imaging, and therapeutic evaluation indicated that both ^213^Bi, a rapidly decaying, high-energy alpha emitter, and ^177^Lu, a longer-lived beta-emitter, were effective in the treatment of this aggressive melanoma model (Figure 1a). Side by side comparison of the two isotopes indicated that treatment with ^213^Bi-h8C3 provided pronounced delay in the tumor growth rate with a significant dose response observed for the 14.8 MBq vs. 7.4 MBq doses, with no adverse effects (Figure 1b). Treatment with ^177^Lu-h8C3, although effective at reducing the tumor growth rate, showed no dose-dependence improvement, and presented measurable and prolonged adverse hematological toxicity (Figure 1c). Overall, in an aggressive B16-F10 murine melanoma model, ^213^Bi-h8C3 proved to be more effective, and presented no adverse effects [49], which could be due to the higher relative biological effectiveness (RBE) and short half-life of ^213^Bi.

Transition into the syngeneic S91 melanoma in DBA/2 mice allowed us to evaluate the utility of ^213^Bi-h8C3 RIT in combination with anti-PD-1 immunotherapy [50]. Combination therapy resulted in significant delay of tumor development and improved survival, relative to monotherapy. These results establish syngeneic S91 melanoma in DBA/2 mice as an ideal model for the pre-clinical evaluation of anti-melanin RIT in combination with immunotherapy. Further evaluation in this model of a long-lived alpha-emitter, ^225^Ac, versus beta-emitter ^177^Lu, in combination with immune checkpoint blockade (ICB), indicated a synergistic effect, although this effect could not be linked to the infiltration of T cells into the tumor microenvironment (TME) [51]. This could be due to the temporal nature of these events. In conclusion, ^177^Lu, with its recent clinical successes such as Lutathera^®^, proved to be ideal in this melanoma model (S91/DBA2), working in concert with ICB, resulting in the synergistic reduction in tumor growth and improved survival [51]. Our work includes the development of a humanized anti-melanin antibody that is effective at targeting melanoma in mouse models, the use of this mAb with clinically relevant isotopes for treatment of experimental melanoma with RIT, and co-treatment with ICB with indications of synergistic effect. Taken together, this pre-clinical work positions melanin-targeting RIT for transition into clinical evaluation.

## 4. Targeting Melanin with Small Molecules

### 4.1. Benzamides

One approach utilizing small molecules is exploiting the ability of benzamides (Figure 2A) to target intracellular melanin. Due to their lipophilicity, they are able to efficiently penetrate the cellular membrane, contrary to the antibody approach of targeting extracellular melanin. Initially positive results have shown the remarkable ability of these benzamides to specifically target tumors and be retained there. Labarre et al. investigated the mechanisms of benzamides binding to melanin [52]. They concluded that there are two classes of binding sites that contribute to benzamide affinity for melanin—one type depends on electrostatic forces and the other involves hydrophobic interactions [52]. Due to these properties, initial tests as diagnostic tools [53,54,55] have led to the evaluation of benzamides as therapeutics.

A theranostic clinical study was conducted in 2014 utilizing a radio-iodinated benzamide compound I-BA52, which was initially radiolabeled with ^123^I for diagnosis followed by ^131^I for therapy [56]. Patients were pre-selected using histologically confirmed metastatic melanoma and given ^123^I-BA52 for planar and SPECT/CT imaging (twenty-six patients in total). The imaging revealed that there was a low transient uptake in the excretory organs and high retention in the tumor, agreeing with the results of a previous study utilizing the radio-iodinated benzamide compound MIP-1145 [57]. Of these twenty-six initial patients, nine were selected to receive a therapeutic dose of ^131^I-BA52. The therapeutic doses of ^131^I-BA52 were well-tolerated and deemed safe with no acute or mid-term toxicities. The dose of 12.2 Gy/GBq was delivered to the tumor and the highest off-target dose of 3.1 Gy/GBq was delivered to the lungs. Though follow-up studies are needed to determine the ideal dose, the initial results gave measurable anti-tumor effects (Figure 3), indicating the exciting promise and efficacy of these compounds [56].

Other groups have been also investigating benzamide compounds as possible therapeutic agents for melanoma [58,59,60,61]. A recent review described the pre-clinical use of benzamides as imaging agents including multi-modality and therapeutic agents [62]. Among the latter, a radioiodinated *N*-(2-diethylaminoethyl)-6-iodoquinoxaline-2-carboxamide dihydrochloride salt, abbreviated [^125/131^I]ICF01012 or [^125/131^I]3, which is a quinoxaline benzamide (BZA) derivative, has been investigated most thoroughly. Degoul et al. [60] demonstrated that [^131^I]ICF01012 produced a significant anti-tumor effect by decreasing the primary tumor growth and dissemination process in a B16-BL6 melanoma model. In this regard, a single administration of [^131^I]ICF01012 (in the range of 14.8–22.2 MBq) resulted in decreased tumor growth and only transient hematological toxicity. The important safety concerns of the treatment with radiolabeled benzamides are healthy pigmented tissues. In this study, no damage of the cutaneous melanocytes and skin was observed. When the retina of the eyes was examined histologically in the treated mice, no alterations were recorded in 30% of the treated mice while in the remaining 70% of the animals, the damage was only seen in the optic nerve area. The dosimetry calculations performed with the help of the medical internal radiation dose (MIRD) methodology revealed that the absorbed dose to the main organs was low (<4 Gy) while 30 Gy was delivered to the tumor. The histological analysis of the tumors showed that this radiation dose initiated robust radiobiological effects, which resulted in decreased proliferation and survival of the tumor cells and decreased angiogenesis. Moreover, an increase in the tumor suppressor gene expression, melanogenesis, and anti-angiogenic markers was observed. The authors attributed their observations to the tumor cell death mechanism occurring via mitotic catastrophe. These results provided the impetus for the further investigation of [^131^I]ICF01012 as a leading pre-clinical candidate for the treatment of metastatic melanoma.

Radiation dosimetry is very important for moving a radiopharmaceutical candidate along the translational path toward a clinical trial. Viallard et al. [63] performed dosimetry estimation for two melanoma models—pigmented murine B16F0 and human SK-Mel3. Using in vivo imaging, they observed the uptake of [^123^I]ICF01012 in the melanoma tumors. The radiation dose to the tumor was directly proportional to the tumor’s melanin content. Fractionated treatment with 3 × 25 MBq of [^131^I]ICF01012 delivered 53.2 Gy to the tumor significantly slowed down the SK-Mel 3 tumor growth and resulted in the prolongation of the median survival in treated animals. Jourberton et al. [64] took the dosimetry evaluation of [^131^I]ICF01012 further by performing the SPECT/CT and ex vivo measurements in healthy rabbits. The doses delivered to the eyes and liver were dose-limiting with 45.8 ± 7.9 Gy/GBq and 6.38 ± 0.50 Gy/GBq, respectively. However, conversion of the former into the dose to the human retina resulted in a significantly lower value of 3.07 ± 0.70 Gy/GBq. The authors concluded that [^131^I]ICF01012 was a promising candidate for a clinical trial and for a personalized dosimetry approach.

The pre-clinical work on evaluating the efficacy and safety of [^131^I]ICF01012 is summarized in Table 1. It culminated in the clinical trial NCT03784625 being initiated in 2019 [65]. According to clinicaltrials.gov:

“the study will include a maximum of 36 patients. This study will begin with a preselection part that consists of an injection of [^131^I]ICF01012 at a diagnostic dose (185 MBq) in order to preselect patients who will receive the therapeutic dose according to the dosimetry results: binding of [^131^I]ICF01012 on at least a tumoral lesion and an acceptable radiation absorbed dose to major organs. The second phase will consist of a therapeutic part with a single administration of [^131^I]ICF01012 at a therapeutic dose. This part is a dose escalation model (4 levels of therapeutic dose to be tested)”.

The efficacy and the safety results of this trial, especially toward healthy melanized tissues such as the retina of the eye, are highly anticipated.

Modern oncology rarely uses a single drug for a metastatic cancer treatment, and very often relies on a combination of treatments. In addition, mechanistic studies are crucial to find a combination of drugs that can affect the tumor via tangential mechanisms. Viallard et al. [66] investigated the combination of [^131^I]ICF01012 with coDbait, a DNA repair inhibitor. They observed the synergy of coDbait and [^131^I]ICF01012 in an SK-Mel3 melanoma xenograft, and an additive effect in the murine syngeneic Bl6-F10 model toward slowing down the tumor growth. This was observed with coDbait not increasing the side effects of radionuclide therapy in healthy melanized tissues such as hair follicles and the eyes. From a mechanistic standpoint, DNA strand breaks from radionuclide therapy were not enhanced by coDbait. However, the findings of micronuclei and cell cycle blockade in treated tumors point to coDbait interfering with DNA repair. Akil et al. [67] demonstrated both in vitro and in vivo the ability of [^131^I]ICF01012 to decrease the metastatic capacity of melanoma cells via epithelial–mesenchymal transition-like (EMT-like) reduction and induction of cell differentiation. The same group examined the interaction between [^131^I]ICF01012 and the immune system [68]. This topic is important as immunotherapy with checkpoint inhibitors is currently a standard of care for metastatic melanoma. The treatment of tumor-bearing immunocompetent mice with the combination of [^131^I]ICF01012 and anti-CTLA-4, anti-PD-1, anti-PD-L1 ICBs resulted in prolonged survival above either drug alone. The mechanistic experiments revealed that the tolerance was a dominant immune escape mechanism while exhaustion was not detected after treatment with [^131^I]ICF01012. Finally, the same researchers studied therapy with [^131^I]ICF01012 alone or in combination with MEK inhibitors (MEKi) [69]. They utilized spheroidal and in vivo melanoma models that harbor constitutive MAPK/ERK activation thought to be responsible for tumor radioresistance. Taken together, their results provide the evidence that the [^131^I]ICF01012 and MEKi combination can be beneficial for the treatment of the advanced pigmented BRAF-mutant melanoma and that [^131^I]ICF01012 alone is promising for the treatment of NRAS-mutant melanoma.

### 4.2. Nicotinamides and Picolinamides

Nicotinamides (Figure 2B) and picolinamides (Figure 2C) are another class of small molecules capable of binding melanin, which have been investigated as potential radiotherapeutics for metastatic melanoma. Chang et al. [70] reported synthesis, in vitro and in vivo evaluation in mouse models of a novel (^123^/^131^)I-labeled nicotinamide derivative that specifically binds to melanin, (^123^/^131^)I-Iochlonicotinamide. This compound has been proven to be quite stable in vitro and demonstrated high uptake into melanotic B16-F0 melanoma cells in contrast to A375 amelanotic cells. When administered to tumor-bearing mice, (^123^/^131^)I-Iochlo-nicotinamide showed high uptake in melanotic tumors and a high tumor-to-muscle ratio. The researchers also performed microSPECT imaging and dosimetry calculations with the projected radiation-absorbed dose for a human being of approximately 0.44 mSv/MBq. They concluded that (^123^/^131^)I-Iochlonicotinamide could be further developed as a promising theranostic agent for treating malignant melanoma.

Xu et al. directed their effort toward synthesis and the in vitro and in vivo evaluation of ^131^I-N-(2-(diethylamino)ethyl)-5-(iodo-^131^I)picolinamide (^131^I-5-IPN) [71]. The pharmacokinetics of ^131^I-5-IPN were assessed via SPECT and biodistribution in the B16-F10 tumor models and in A375 xenografts, and targeted therapy was performed in B16-F10 melanoma-bearing mice. 31I-5-IPN displayed high tumor uptake in B16-F10 tumors for up to 72 h, with a dose of 18.5 MBq extending the median survival compared to the control groups. Analyses of the biomarkers post-treatment with ^131^I-5-IPN revealed the reduced expression of proliferating cell nuclear antigen (PCNA) and Ki67 as well as the cell cycle blockage in the G2/M phase in tumor tissues. In addition, decreased vascular endothelial growth factor and CD31 expression pointed to the decrease in the tumor growth.

Finally, Chen et al. attempted to synthesize melanin-binding molecules that would combine the structural motifs of picolinamides, nicotinamides, and benzamides in their structure [72]. To achieve this, they conjugated picolinamide/nicotinamide with the pharmacophore of ^131^I-MIP-1145 to obtain ^131^I-iodofluoropicolinamide benzamide (^131^I-IFPABZA) and ^131^I-iodofluoronicotiamide benzamide (^131^I-IFNABZA). Biological side-by-side comparison of these two molecules in the B16-F10 tumor-bearing mice showed that ^131^I-IFNABZA exhibited a higher tumor-to-muscle ratio than ^131^I-IFPABZA. Both tracers demonstrated low tumor uptake in the A375 amelanotic melanoma-bearing mice, attesting to their high specificity for melanin (Figure 4). When compared to the literature data on radiation doses delivered to the tumors by other melanin-binding radiotherapeutics, the radiation-absorbed dose to the tumor from ^131^I-IFNABZA was higher, while the doses to the normal tissues were lower. The authors concluded that the ^131^I-IFNABZA combinatorial molecule has potential as a theranostic agent for metastatic melanoma.

## 5. Conclusions

Recent developments of radionuclide therapy targeting melanin have shown that different approaches such as melanin-targeting antibodies and benzamides could both achieve success in clinical trials. Other types of melanin-binding small molecules are still in the pre-clinical stage. Both radiolabeled antibodies and radiobenzamides worked in concert with the immunotherapy of melanoma via a variety of immune mechanisms in the pre-clinical studies, showing a potential for combination therapies in the future.

## 6. Future Perspectives

Radionuclide therapy, in general, is more cost effective than immunotherapy and has less side effects, which can make it competitive with the existing or developing immunotherapies. Importantly, the imaging of patients with melanin-binding compounds or antibodies radiolabeled with PET- or SPECT-enabling radionuclides will allow us to pre-select patients whose tumors’ high uptake of the targeting molecules makes them candidates for radionuclide therapy. The second-generation of radiolabeled benzamides, which is currently in clinical trials, seems to be the most promising in this regard. Further clinical work will clarify whether the “double-edged sword” nature of melanin being simultaneously a therapeutic target and a radioprotector in melanoma will allow the radionuclide therapy to result in increasing the overall survival of melanoma patients. It is our hope that in the not too distant future, radionuclide therapy will become a standard of care for patients with metastatic melanoma.

## Figures and Tables

**Figure 1 ijms-23-09520-f001:**
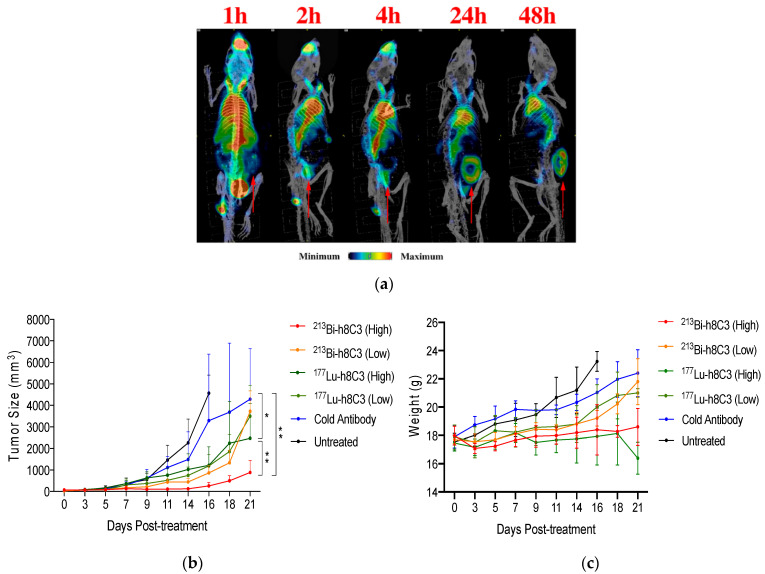
The radioimmunoimaging and radioimmunotherapy of murine melanoma with h8C3 mAb to melanin. (**a**) The 111In-h8C3 microSPECT/CT maximum intensity projection (MIP) time course imaging study (tumor location indicated by arrows). (**b**) Tumor size (volume) and (**c**) mouse weight over the course of the study with high dose being 14.8 MBq, and low dose being 7.4 MBq of the respective isotopes. Stars indicate statistical significance: * indicates *p* value smaller than 0.05 and ** indicates *p* value smaller than 0.01. (adapted from ref [49]).

**Figure 2 ijms-23-09520-f002:**
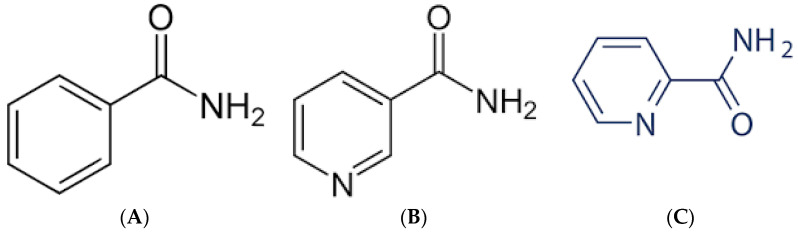
The basic structures of benzamide (**A**), nicotinamide (**B**), and picolinamide (**C**).

**Figure 3 ijms-23-09520-f003:**
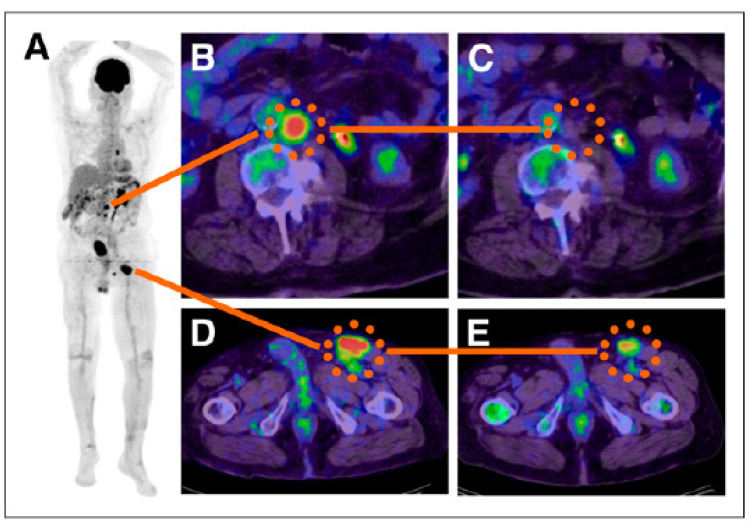
The efficacy of treating metastatic melanoma with I-BA52. A patient (59 year-old) with malignant melanoma underwent pre-therapeutic 18F-FDG PET to identify the extent of metastases (**A**). A comparison of the standardized uptake values (SUV) of pre-therapeutic metastatic tumors (**B**,**D**) to 6 week follow-up revealed a significant decrease in SUV (**C**,**E**) with one example showing 18F-FDG-negative post therapy (**C**). The red color indicates the highest uptake and blue color the lowest. (adapted from [56]).

**Figure 4 ijms-23-09520-f004:**
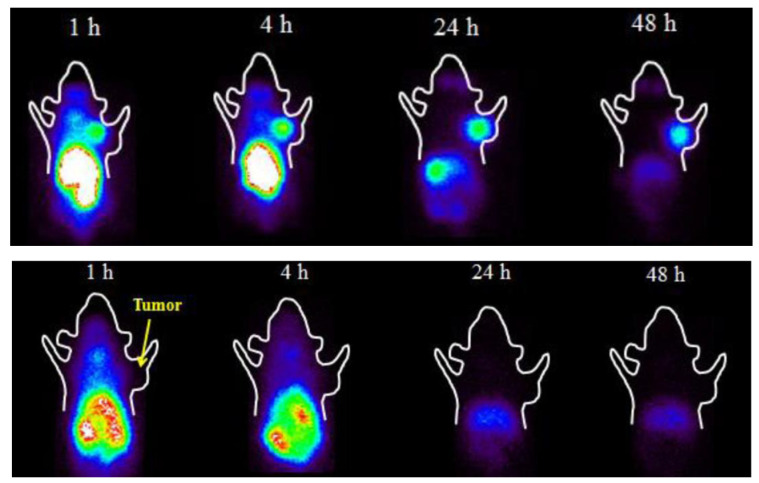
The biological evaluation of ^131^I-iodofluoropicolinamide benzamide (^131^I-IFPABZA) in B16-F10 melanotic tumor-bearing mice (upper row) and in A375 amelanotic melanoma-bearing mice (lower row). The tracer demonstrated a high uptake in B16-F10 tumors and low tumor uptake in A375 tumors, attesting to its high specificity for melanin (adapted from [72]).

**Table 1 ijms-23-09520-t001:** Important results in developing radiolabeled benzamides for melanoma therapy.

Studies and Results	Reference
First in man study of ^131^I-BA52. The dose of 12.2 Gy/GBq was delivered to the tumor and the highest off-target dose was 3.1 Gy/GBq, the drug was well-tolerated and deemed safe with no acute or mid-term toxicities.	[56]
[^131^I]ICF01012 administration in the range of 14.8–22.2 MBq resulted in decreased tumor growth and only transient hematological toxicity in B16-BL6 melanoma mice. No damage to the retina of an eye was recorded in 30% of the treated mice while in the remaining 70% of the animals, the damage was only seen in the optic nerve area.	[60]
Radiation dosimetry study of [^123^I]ICF01012 in murine melanoma showed that fractionated treatment with 3 × 25 MBq of [^131^I]ICF01012 delivered 53.2 Gy to the tumor.	[63]
Further radiation dosimetry evaluation of [^131^I]ICF01012 was conducted by performing the SPECT/CT and ex vivo measurements in healthy rabbits. The doses delivered to the eyes and liver were dose-limiting with 45.8 ± 7.9 Gy/GBq and 6.38 ± 0.50 Gy/GBq, respectively. However, the conversion of the former into the dose to the human retina resulted in a significantly lower value of 3.07 ± 0.70 Gy/GBq.	[64]
On-going clinical trial NCT03784625, which was initiated in 2019 [65]. According to clinicaltrials.gov: “the study will include a maximum of 36 patients. This study will begin with a preselection part that consists of an injection of [^131^I]ICF01012 at a diagnostic dose (185 MBq) in order to preselect patients who will receive the therapeutic dose according to the dosimetry results: binding of [^131^I]ICF01012 on at least a tumoral lesion and an acceptable radiation absorbed dose to major organs. The second phase will consist of a therapeutic part with a single administration of [^131^I]ICF01012 at a therapeutic dose. This part is a dose escalation model (4 levels of therapeutic dose to be tested)”.	[65]

## Data Availability

Not applicable.

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
