# Peer review of "Targeting Melanin in Melanoma with Radionuclide Therapy"

_ijms, 2022, doi:10.3390/ijms23179520_

Round 1

Reviewer 1 Report

The presented manuscript reviews and summarizes the results of the experiments on radionuclide therapy of melanoma, with melanin as a target. The manuscript is well-written and organized and concerns a very interesting problem. In my opinion the manuscript is worth publishing after some corrections, which are as follows:

- the article lacks the methodology information, i.e. what databases were searched, whata were the key words during the search, whata were the exclusion criteria, in any, etc. This part should not be missed in review-type articles

- apart from conclusions, the authors are encouraged to draw some future perspectives for the reviewed therapeutic approach, i.e. what are the most promising compounds among those that were reviewed in the manuscript etc. 

Author Response

The presented manuscript reviews and summarizes the results of the experiments on radionuclide therapy of melanoma, with melanin as a target. The manuscript is well-written and organized and concerns a very interesting problem. – Response: We would like to thank the Reviewer for his/her encouraging opinion about our work.

In my opinion the manuscript is worth publishing after some corrections, which are as follows:

- the article lacks the methodology information, i.e. what databases were searched, what were the key words during the search, what were the exclusion criteria, in any, etc. This part should not be missed in review-type articles – Response: The following information has now been included into the revised manuscript under the newly created  Methodology section: “The PubMed and Web of Science databases were searched, with years of search being 2011-2022. The search terms were: melanoma, targeted radionuclide therapy, melanin, radioimmunotherapy and clinical trials and they were used in various combinations. The exclusion term was α-melanocyte-stimulating hormone (α-MSH). The older than 2011 references were included only if required to provide the background information on melanin or targeting molecules.”

- apart from conclusions, the authors are encouraged to draw some future perspectives for the reviewed therapeutic approach, i.e. what are the most promising compounds among those that were reviewed in the manuscript etc. – Response: We have modified the Conclusions section and have created the Future perspective section in the revised manuscript which reads: “ 5.0. Conclusions Recent developments of radionuclide therapy targeting melanin have shown that such different approaches as melanin-targeting antibodies and benzamides could both achieve success in clinical trials. Other types of melanin-binding small molecules are still in pre-clinical stage. Both radiolabeled antibodies and radiobenzamides have worked in concert with immunotherapy of melanoma via a variety of immune mechanisms in the pre-clinical studies showing potential for combination therapies in the future.  6.0 Future Perspectives Radionuclide therapy in general is more cost effective than immunotherapy and has less side effects which can make it competitive with the existing or developing immuunotherapies. Importantly, imaging of patients with melanin-binding compounds or antibodies radiolabeled with PET- or SPECT-enabling radionuclides will allow to pre-select the patients whose tumors high uptake of the targeting molecules makes them candidates for radionuclide therapy. Second generation of radiolabeled benzamides which is currently in clinical trials seems to be the most promising in this regard. Further clinical work will clarify if the “double-edge sword” nature of melanin being simultaneously a therapeutic target and a radioprotector in melanoma will allow the radionuclide therapy to result in increasing overall survival of melanoma patients. It is our hope that in the not too distant future radionuclide therapy will become a standard of care for patients with metastatic melanoma.“

Reviewer 2 Report

Review on manuscript “Targeting Melanin in Melanoma with Radionuclide Therapy” written by Kevin J.H. Allen et al.

Melanoma is a type of skin cancer, more aggressive being also considered the deadliest of skin cancers. Therefore, the research in this field is trying to overcome the poor survival statistics. Many approaches like targeted therapy, immunotherapy or combined therapy were developed over the years and tried to offer a better outcome for the melanoma-diagnosed individuals but there is still room for improvement.

This manuscript presents a different approach that targets melanin pigment using radionuclide therapy.

Even do the study is interesting I will kindly the authors to take into account the followings

The introduction part is ok but the scope of the review should be outlined and justified.

3.1. Benzamides

- Please add the motivation for which the benzamides binds with melanin.

- Please add a table with summarized important data

- Please add the chemical structure of benzamides – as general, or mention the one used in this study

3.2. Nicotinamides and picolinamides. 

- Please add some details about the nicotinamides and picolinamides, like utilization, chemical structure.

- The subsection seems a little short.

The design- I can understand the use of existing figures from literature but I think that the authors should elaborate at least one figure/scheme as an original contribution

Please motivate why were investigated only these small molecules (benzamides, nicotinamides and picolamides). From my point of view, the study is incomplete and does not fully present the progress made in the last ten years in this field.

Also, please explain how was conducted the selection of the used references.

The references should be written according to the journal format.

Author Response

Melanoma is a type of skin cancer, more aggressive being also considered the deadliest of skin cancers. Therefore, the research in this field is trying to overcome the poor survival statistics. Many approaches like targeted therapy, immunotherapy or combined therapy were developed over the years and tried to offer a better outcome for the melanoma-diagnosed individuals but there is still room for improvement. This manuscript presents a different approach that targets melanin pigment using radionuclide therapy. – Response: We would like to thank the Reviewer for his/her encouraging opinion about our work.

Even though the study is interesting I will kindly the authors to take into account the followings:

The introduction part is ok but the scope of the review should be outlined and justified. – Response: We have clarified the scope of the review in the revised manuscript to make sure that it is not a systematic review.  It now reads: “The scope of this review is to highlight the most promising developments in the last 10 years in the radionuclide treatment of melanoma targeting melanin”. In addition, we have added a Methodology section to the revised manuscript which reads: “under the newly created  Methodology section: “The PubMed and Web of Science databases were searched, with years of search being 2011-2022. The search terms were: melanoma, targeted radionuclide therapy, melanin, radioimmunotherapy and clinical trials and they were used in various combinations. The exclusion term was α-melanocyte-stimulating hormone (α-MSH). The older than 2011 references were included only if required to provide the background information on melanin or targeting molecules.”

3.1. Benzamides

- Please add the motivation for which the benzamides binds with melanin. – Response: We have added the following information and the new reference 52 to the first paragraph of this section: “Labarre et al. investigated the mechanisms of benzamides binding to melanin [52]. They concluded that there are two classes of binding sites which contribute to benzamides affinity for melanin – one type depends on electrostatic forces and the other involves hydrophobic interactions [52].” 

- Please add a table with summarized important data. – Response: We have summarized the important benzamides data in Table 1 in the revised manuscript.

Table 1 Important results in developing radiolabeled benzamides for melanoma therapy

                                                         Studies and Results

 Reference

First in man study of 131I-BA52.  The dose of 12.2 Gy/GBq was delivered to the tumor and the highest off-target dose was 3.1 Gy/GBq, the drug  was well tolerated and deemed safe with no acute or mid-term toxicities.

56

[131I]ICF01012 administration in the range of 14.8 - 22.2 MBq resulted in decreased tumor growth and only transient hematological toxicity in B16-BL6 melanoma mice. No damage to the retina of an eye was recorded in 30% of treated mice while in the remaining 70% of the animals the damage was only seen in the optic nerve area.

60

Radiation dosimetry study of [123I]ICF01012 in murine melanoma showed that  fractionated treatment with 3 × 25 MBq of [131I]ICF01012 delivered 53.2 Gy to the tumor.

63

Radiation dosimetry evaluation of [131I]ICF01012 further by performing the SPECT/CT and ex vivo measurements in healthy rabbits. The doses delivered to the eyes and liver were dose-limiting with 45.8 ± 7.9 Gy/GBq and 6.38 ± 0.50 Gy/GBq, respectively. However, conversion of the former into the dose to human retina resulted in a significantly lower value of 3.07 ± 0.70 Gy/GBq.

64

On-going clinical trial NCT03784625 which was initiated in 2019 [65]. According to clinicaltrials.gov: “the study will include a maximum of 36 patients. This study will begin with a preselection part that consists of an injection of [131I]ICF01012 at a diagnostic dose (185 MBq) in order to preselect patients who will receive the therapeutic dose according to the dosimetry results: binding of [131I]ICF01012 on at least a tumoral lesion and an acceptable radiation absorbed dose to major organs. The second phase will consist of a therapeutic part with a single administration of [131I]ICF01012 at a therapeutic dose. This part is a dose escalation model (4 levels of therapeutic dose to be  tested)”.

65

- Please add the chemical structure of benzamides – as general, or mention the one used in this study. – Response: We have added a new Fig. 2 to the revised manuscript which shows the structures of benzamide, nicotinamide and picolinamide.

3.2. Nicotinamides and picolinamides. 

- Please add some details about the nicotinamides and picolinamides, like utilization, chemical structure. –Response: We have added a new Fig. 2 to the revised manuscript which shows the structures of   nicotinamide and picolinamide.

- The subsection seems a little short. – Response: As stated in the Introduction, this review is not a systematic review and aims to highlights some representative interesting publications.

The design- I can understand the use of existing figures from literature but I think that the authors should elaborate at least one figure/scheme as an original contribution. – Response: We have added Fig. 4 to the revised manuscript which has been adapted from ref. [72].

Please motivate why were investigated only these small molecules (benzamides, nicotinamides and picolamides). From my point of view, the study is incomplete and does not fully present the progress made in the last ten years in this field. – Response: Please see the response to the question about the scope of the review.

Also, please explain how was conducted the selection of the used references. – Response: Please see the newly added Methodology section.

The references should be written according to the journal format. – Response: The references have been reformatted into the IJMS format.

Round 2

Reviewer 2 Report

I appreciate that the authors answered all the queries, at this point the manuscript is clearer and improved.

I congratulate the authors for their work.